

# Role of the novel *aloe vera*-based titanium dioxide bleaching gel on the strength and mineral content of the human tooth enamel with respect to age

Afsheen Mansoor[1,2], Emaan Mansoor[3], Atta Ullah Shah[4], Uzma Asjad[4], Zohaib Khurshid[5] and Amir Isam Omer Ibrahim[6]

[1] Microbiology and Nanotechnology department, Quaid-i-Azam University, Islamabad, Pakistan
[2] Department of Dental Material Sciences, School of Dentistry, Shaheed Zulfiqar Ali Bhutto Medical University, Islamabad, Pakistan
[3] Islamic International Dental College, Riphah International University, Islamabad, Pakistan
[4] Department of Materials, National Institute of Lasers and Optronics, Islamabad, Pakistan
[5] Department of Prosthodontics and Dental Implantology, College of Dentistry King Faisal University, Al Ahsa, Saudi Arabia
[6] Department of Restorative Dental Sciences, College of Dentistry, King Faisal University, Al-Ahsa, Saudi Arabia

Corresponding authors
Afsheen Mansoor,
drafsheenqamar@gmail.com,
drafsheen@szabmu.edu.pk
Amir Isam Omer Ibrahim,
dr.aio.ibrahim@gmail.com,
aiibrahim@kfu.edu.sa

## ABSTRACT

There has been an increased demand for dental bleaching globally irrespective of age and gender. Main drawbacks associated with conventional tooth bleaching agents have been compromised strength and mineral-content of tooth enamel which results in sensitivity, discomfort, roughness, and structure loss of human teeth. Currently, nanoparticles synthesized by green synthesis have gained popularity especially in medical and dental applications because of their versatile and beneficial nano-scaled features. Titanium dioxide nanoparticles ($TiO_2$ Nps) in this study were prepared from green ecofriendly source using the aloe vera plant extract and were then characterized *via* dynamic light scattering (DLS), scanning electron microscope (SEM), X-ray diffraction spectroscopy (XRD), and energy dispersive X-ray (EDX), for size, shape, composition and true-phase. These $TiO_2$ Nps were incorporated in commercial bleaching gel containing hydrogen peroxide to form a novel $TiO_2$-bleaching gel which was used to bleach extracted anterior teeth belonging to four different age groups: 20–29 years, 30–39 years, 40–49 years and ≥50 years. These teeth were investigated for micro-hardness (Vickers microhardness tester) and mineral-content (EDX spectroscopy) including sodium, magnesium, phosphorus, calcium in an *in-vitro* environment both before and after bleaching. Results revealed that $TiO_2$ Nps prepared by aloe vera plant were nanos-sized of about 37.91–49 nm, spherical shape, true anatase phase with pure titanium and oxygen in their composition. The values of Vickers micro-hardness and mineral-content (Na, Mg, P, Ca) of enamel specimens belonging to different age groups enhanced in a linear pattern before bleaching with the increase in age ($p$ value < 0.05). There was negligible reduction observed in Vickers micro-hardness and mineral-content elements (Na, Mg, P, Ca) of all enamel specimens belonging to different ages after the bleaching ($p$ value > 0.05). The novel $TiO_2$-bleaching gel prepared was effective

enough in preventing the declination in Vickers micro-hardness strength and mineral-content of all the enamel specimens belonging to different age groups even after the bleaching procedure which makes it a promising biomaterial.

## INTRODUCTION

For the past few years, tooth bleaching has been the most desired dental treatment encompassing the esthetics globally. However, loss of mineral content from the structure of dental enamel and reduction in its strength after using the traditional hydrogen peroxide bleaching agent in different concentrations has become a point of concern to a larger extent (*Goldberg, Grootveld & Lynch, 2010*). Dental enamel is the hardest, toughest and strongest tissue in the human body that consists mainly of 96% inorganic material, 1% organic material and 3% water by weight. The inorganic material in enamel is predominantly calcium and phosphorus basically in the form of hydroxyapatite $Ca_{10}(PO_4)6(OH)_2$ that contributes in its crystalline structure mostly (*He et al., 2011*). Magnesium (Mg), and sodium (Na) are other important elements found in the enamel's structure while carbon (C), fluoride (F), potassium (K), zinc (Zn), lead (Pb), nitrogen (N), and iron (Fe) are present in additional amounts as trace elements (*He et al., 2011*).

Normal aging process has been well known for causing major changes in the human body, especially dentition such as color, size, shape, texture, mechanical strength and mineral. Young human teeth has yellowish tinge that becomes darker with the slow and gradual progression in age. On the other hand, micro-hardness and mineral content of the tooth enamel has displayed an increase in a systematically linear pattern with aging (*Pinto et al., 2004*). This linear relationship between hardness and mineral content of the tooth enamel has been observed in the elements of sodium (Na), magnesium (Mg), calcium (Ca), and phosphorus (P). Hence, it has been confirmed that old age human tooth enamel has revealed enhanced micro-hardness with highly mineralised structure as compared to the young tooth enamel (*He et al., 2011*; *Mansoor et al., 2022b*).

Specific age related changes in micro hardness and elemental composition of human teeth were first reported in 1889 by *Lacassagne, Sabouraud & Moreau (1889)*. They divided the age related changes into two well-differentiated stages referred to as the initial (first) and final (secondary) stages. The initial stage has allowed the slightly observed variations in the texture, structure, morphology, and mineral content of the human tooth enamel at the time of formation (*Murray et al., 2002*). The final stage includes the secondary changes that occur after the formation and eruption of the permanent human tooth. These common changes with aging were attrition, abrasion, wear, gingival recession, root transparency, root resorption, apposition of secondary cementum, mineralization, de-mineralization, re-mineralization, and sometimes degeneration.

Several other studies have reported an increase in the micro-hardness, mineral content, and dentin thickness altogether with the color change getting darker and darker with the

advancement in age (*Mansoor et al., 2022b*; *Gustafson, 1950*; *Mansoor et al., 2019*). Another research conducted recently has also narrated the direct correlation between the strength and composition of the human tooth with aging. This study concluded the significant increase in micro-hardness and mineral content of the human teeth with the advancement in age (*Mansoor et al., 2020*). Thus, it has been demonstrated that these variables have been taken as the best tools for estimating the exact age of any human being (*Mansoor et al., 2022b*; *Gustafson, 1950*; *Mansoor et al., 2019*).

Furthermore, despite the complete maturation of the human teeth, the secondary changes have been found to continue throughout the life of any individual even after the maturation. Such changes include the increase in the permeability and the porosity of the human tooth enamel (*Murray et al., 2002*). The young tooth enamel has depicted more enhanced permeability and porosity at the time of eruption before maturation. As a result, this highly permeable and porous tooth enamel might get more susceptible to the process of de-mineralization which becomes a strong drawback of youth (*Park et al., 2008*). This in turn would lead to the declination of the micro-hardness, and mineral content of the young tooth. With maturation, the human tooth has shown less permeability and porosity in its structure thus, reducing the chance of getting de-mineralized in a limited manner. This demineralization in the old aged tooth enamel can not be stopped completely because of the environmental factors also (*He et al., 2011*). These age related changes might have a strong implication on the human tooth color also that needs to be investigated.

Dental bleaching has been the only recommended treatment option for managing the tooth discoloration. This treatment option is commonly used by professional dentists to convert the discolored teeth into brighter and lighter ones *via* commercially available dental bleaching agents. The most commonly used conventional bleaching agents consisted of hydrogen-peroxide in concentrations ranging between 35–38%. It has been concluded that these conventional bleaching agents (35–38%) have shown significant improvement in the color of teeth but declined their micro-hardness and mineral content (*Alqahtani, 2014*; *Berger et al., 2010a*). Previously, it has been demonstrated that hydrogen peroxide in a conventional bleaching agent has reduced the enamel and dentine shear bond strength, micro-hardness (*Berger et al., 2010a*), and phosphate and calcium concentrations (*Berger et al., 2010b*). It has been concluded that the obvious differences in the peroxide type, peroxide-concentration, bleaching- protocol, and evaluating method is responsible for inducing the adverse effects on the human tooth and its structures including enamel and dentine. The side effects associated with conventional bleaching agents have been found to be tooth sensitivity and discomfort that might have occurred due to loss of hardness and mineral content from the bleached human tooth structure. Therefore, further investigations have been required to improve the color of teeth without compromising their hardness and mineral content (*Smidt, Feuerstein & Topel, 2011*).

The utilization of nanotechnology in various dental and medical fields became possible because of its multiple versatile and advanced characteristics which incorporates their magnetic, physical, chemical, mechanical, strength and hardness properties. The metal-oxide nanoparticles *e.g.,* Fe, Si, Al, Ti, Se, P, Co, Ag, Au, S, Mg, Cd, Cu, Pd, Sn, Pb and Zn are most commonly used in various significant applications. Currently, titanium dioxide

nanoparticles (TiO$_2$ Nps) have gained popularity because of their unusually versatile properties such as: wear resistance, low thermal conductivity, scratch resistance, corrosion resistance, high electrical conductivity, and high thermal diffusivity, (*Long et al., 2014*). Additionally, TiO$_2$ Nps is lightweight, cost-effective, and fatigue resistant displaying the least level of allergy and toxicity (*Jorge et al., 2013*; *Mansoor et al., 2022c*) with excellent antimicrobial, anti-parasitic, and anti-inflammatory properties (*Niinomi, 2008*). Due to its biological compatibility with human tissues, TiO$_2$ Nps have been used in various dental applications (*Mansoor et al., 2022b*).

A researcher concluded that the strength, hardness, physical, chemical, mechanical, and biological properties of a restorative material has been enhanced after the utilization of commercially available TiO$_2$ Nps (*Tahir et al., 2020*) but their biocompatibilty might pose a problem. Therefore, green synthesis involving the fabrication of safe, sustainable, biocompatible, and environmentally friendly materials has been the utmost requirement because of the incorporation of natural biological resources such as plants. The medicinal plants have gained extra importance in the current epic because of their certain beneficial effects when used for human health and safety purposes (*Yuvasree, Nithya & Neelakandeswari, 2013*). Aloe vera is one of the most potent, famous mineral and vitamin producer medicinal plants that have been in common use since 1st century A.D (*Yuvasree, Nithya & Neelakandeswari, 2013*). To date, aloe vera plant has been renowned for its strong detoxifying, bacteriostatic, antiseptic, bactericidal, antiviral, and immune boosting properties along with generating the large amounts of minerals, vitamins, and amino acids required by healthy human beings. This medicinal plant contains large amounts of vitamin-E, vitamin-A, vitamin-B12, vitamin-E, vitamin-C, folic acids, vitamin-B, amino acids and many other minerals required by healthy human body. Thus, aloe vera is a proven and potent natural remedy used for treatment of cuts, burns, insect bites, skin irritations and injuries. This enhances the chance of using this plant in the medical and dental applications (*Rajeswari et al., 2012*). These versatile properties of the aloe vera plant might be useful in amplifying the hardness and mineral content of the human tooth structure if it has been incorporated into the conventional dental bleaching agent.

Therefore, the aim of the current study was to prepare the novel titanium-dioxide nanoparticles *via* aloe vera plant and characterize them, for their physical and chemical properties in the first step. The incorporation of these aloe vera TiO$_2$ Nps into the conventional hydrogen peroxide tooth bleaching gel was carried out to fabricate a novel TiO$_2$-Tooth bleaching gel and then, evaluate its effect on the micro-hardness and mineral-content of tooth enamel of different age groups in second step.

## MATERIALS AND METHODS

### Synthesis and characterization of titanium dioxide nanoparticles

Aloe vera plant was taken from the local garden of Islamabad, Pakistan. A solid mass of 25 g of the aloe vera plant was cut into small pieces, properly ground, and then dissolved into 100 mL of distilled water. The mixture was heated at 90 °C for 2 h in order to get an aloe vera plant extract. The extract was filtered and all the unwanted by-products were removed.

Finally, stock solution containing 1.00 M titanium-chloride (TiCl$_4$, CAS number: 7550-45-0 Alfa Aesar, Thermo Fisher Scientific, Waltham, MA, USA) was prepared by dissolving it in 100 mL of deionized water. The initially prepared aloe vera plant extract was added drop wise in the titanium-chloride (TiCl4) stock solution under constant stirring for 4 h until its pH became 7.0. Finally, crystals of pure titanium dioxide nanoparticles with a chalky white color appeared at the bottom of the mixing flask. The precipitated nanoparticles were washed with ethanol and water in a centrifuge, dried at 100 °C in an oven for one hour, and followed by calcination for 4 h in a furnace at 500 °C (*Kandregula et al., 2015*). The prepared titanium dioxide nanoparticles were then characterized for their size, shape, texture, morphology, surface, composition, and true phase by using the scanning electron microscope (SEM) and energy dispersive X-ray spectroscope (EDS) (NOVA_ Nanosem no_430, FEI-Company, Hills_Boro, OR; USA), dynamic light scattering spectroscope (Zetta-Size Nano- ZSSS Apparatus ZENN 360, Malvernn panalytical Malverrns company, United-Kingdom), and X-ray diffraction pattern spectroscope (XRD) (DP/MAXZ_2400-Diffractometer, Rigaku Corporation, Akishima-Tokyo; Japan) (*Mansoor et al., 2022b*; *Liang et al., 2017*).

## Synthesis of aloe vera based titanium dioxide—hydrogen peroxide bleaching gel

The novel dental bleaching gel was formed through a modified method adapted from the literature (*Hamza et al., 2021*). The 20 weight% of novel aloe vera based titanium dioxide nanoparticles were added into 0.3 g powder (SDI Pola Office bleaching kit, SDI Dental Limited, Dublin, Ireland) containing 35% hydrogen peroxide (2.00 ml). A mixing stirrer was used to mix both powders to get a homogenous mixture of a novel 20% titanium dioxide nanoparticles containing tooth bleaching gel.

## Strength and mineral-content evaluation of novel titanium-dioxide tooth bleaching gel

### Specimen preparation of human tooth enamel

Ethical approval for this study was taken from School of Dentistry, Shaheed Zulfiqar Ali Bhutto Medical University Islamabad, Pakistan with the reference number (SOD/ERB/2022/24). Extracted upper anterior teeth were used in this study, the teeth were extracted due to periodontal involvement and not for the purpose of this study. The inclusion criteria included upper anterior teeth severely affected by periodontal disease, with intact labial surfaces, and no cracks, caries, or fillings. The convenience sampling method used in our study collected 28 extracted central incisors of different age groups ranging between 20-50 years. Any calculus or deposit adhered to the surfaces of these teeth were removed using ultrasonic-scaler and were polished with the pumice powder. The teeth were stored in thymol solution (buffered 0.1% pH 7.00) initially. These teeth were divided into four groups according to their ages 20–29 years ($n = 7$), 30–39 years ($n = 7$), 40–49 years ($n = 7$), and $\geq 50$ years ($n = 7$). The crowns of the teeth were cut from their roots at the cemento-enamel junctions using slow speed cutting-saw (Buehler, isomet panasonic, Tokyo; Japan). The crowns were then cut longitudinally with low speed cutting-saw (Buehler, isomet panasonic, Tokyo; Japan) into several enamel specimen
having thickness of 3.00 mm, width of 4.00 mm and height of 6.00 mm. Enamel specimen ($n = 7$) for each age group was fixed on the phenolic-mould (Kemet-Germany). Initially, each enamel specimen was polished with silicon carbide paper of different grits (1,200, 2,400, and 4,000 grit; Allied High Tech Product Inc, Compton, CA, USA) followed by polishing the specimen with the alumina polishing paste of various particle sizes including 1 µm, 0.3 µm and 0.05 µm (Allied High Tech Product Inc, Compton, CA, USA) in a rotary polishing machine (J.p.s etudes constructions; 27 rue klock, 92 cuchy 737.07.38, M.A; USA) for attaining a highly polished and smooth surface. The enamel specimens were disinfected by keeping them for 10 min in an Ultrasonic bathtub (Ultrasonic Cleaner Branson-I, Smith-Klinee Companys, OR; USA). All enamel specimens were stored in an artificial saliva at 37 °C during the whole investigation for preventing undesired alterations (*Kwon et al., 2015*).

### Bleaching protocol with novel titanium dioxide nanoparticles bleaching gel

All enamel specimens ($n = 7$) belonging to different age groups ranging between 20–50 years involved in this study were bleached by using the synthesized 20% titanium dioxide nanoparticles containing tooth bleaching gel. Bleaching was performed using the novel tooth bleaching gel which was activated using LED-LASER light (Bleaching Lase Light-Plus, DMCⓇ, Odontologica, Sao Carlos SP, Brazil) according to the manufacturer's directions. Total bleaching of 30 min was performed on each enamel specimen after applying the novel bleaching gel on the enamel specimen that belonged to a different group. All enamel specimens were tested for their Vickers micro-hardness and mineral content both before and after carrying out the bleaching protocol. The Vickers micro-hardness and mineral content of enamel specimens before bleaching was taken as control groups while the Vickers micro-hardness and mineral content of enamel specimens after bleaching was considered as experimental groups.

## Micro-hardness evaluation of tooth enamel

Enamel specimens belonging to different age groups ranging between 20–50 years involved in this study were embedded in acrylic poly-cylinders having a width of two cm and a height of three cm. A dimension of four mm × six mm from the labial surfaces of each enamel specimen was left exposed for performing the Vickers micro-hardness testing (*Mansoor et al., 2019*). Each enamel specimen was positioned perpendicular to the long axis of the diamond indenter attached to the Vickers micro-hardness tester (401-MVD, VD-414, Wolpert Gruppe, Bretzfeld, Germany) used in the study for recording the Vickers micro-hardness. A load of 200 g for 15 s was applied to each enamel specimen. Three indentations were applied on each enamel specimen and the distance between each indentation was 100 µm. The average of the three indentations was recorded for all the enamel specimens both before (control) and after (experimental) the bleaching (*Tahir et al., 2020*).

### Mineral-content evaluation of tooth enamel

The Energy dispersive X ray spectroscope (EDX) scan of all the enamel specimens belonging to different age groups was carried out both before (control) and after (experimental) the bleaching procedure with the novel 20% titanium dioxide nanoparticles containing tooth

bleaching gel. The EDX scan was checked for the availability of different elements present in the enamel specimens with the help of a scanning electron microscope (NOVA Nanosem no. 430; FEI-Company, Hillsboro, OR, USA).

### Statistical analysis

IBM SPSS v 20 was used for the statistical analysis of this study. The mean values with standard error and standard deviation for all enamel specimens regarding the Vickers microhardness and presence of different elements were calculated and their significance was obtained by using the one way analysis of variance (ANOVA). *Post hoc* Tukey test was used for the inter-group comparisons among different enamel specimens belonging to 20 years, 30 years, 40 years, and 50 years used in our study by keeping the significance at the level of $p$ value $< 0.05$.

## RESULTS

### Preparation and characterization of titanium dioxide nanoparticles

The presence of white colored precipitates in the mixing flask confirmed the formation of the titanium dioxide nanoparticles. Moreover, the titanium dioxide nanoparticles prepared by the aloe vera plant extract notified the appearance of clustered spherical shape of nanoparticles with the help of scanning electron micrpscopy (Fig. 1).

The XRD scan of the titanium dioxide nanoparticles prepared by the aloe vera plant extract was obtained from the Joint Committee on Powder Diffraction Data (JCPDS) card. This JCPDS data card was known for determining the actual microstructure of the nanoparticles, their composition and phases. This XRD scan revealed the 100% anatase phase in the JCPDS card no: 01-071-1169 at main peak of 2 theta (101) 25.27°, and certain other peaks of (103) 36.87°, (004) 37.69°, (112) 38.50°, (200) 47.98°, (105) 53.76° (211) 54.99°, (204) 62.57°, (116) 68.59°, (220) 70.19° and (301) 75.93°. The particle size calculated by this scan was about 37.91 nm which was obtained by scherrer Debye formula's equation. Thus, X-ray diffraction pattern scan of aloe vera based titanium dioxide nanoparticles displayed prominent peaks of totally pure anatase phase (Fig. 2).

The hydrodynamic size of titanium dioxide nanoparticles prepared by the aloe vera plant extract was found to be 49 nm in diameter. The hydrodynamic size was calculated by dynamic light scattering spectroscopy that showed prediminently sharp peaks (Fig. 3).

The EDS analysis depicted that titanium dioxide nanoparticles prepared by the aloe vera plant extract showed the pure oxygen and titanium peaks in its spectrum. The atomic % of oxygen was 72.87% and weight% was 47.29%. On the other hand, the atomic % of titanium was 27.13% and weight% was 52.71% with no other additional peaks. The absence of any additional peak confirmed the purity of these nanoparticles. Hence, elemental composition of aloe vera based titanium dioxide nanoparticles depicted pure titanium and oxygen peaks only (Fig. 4).

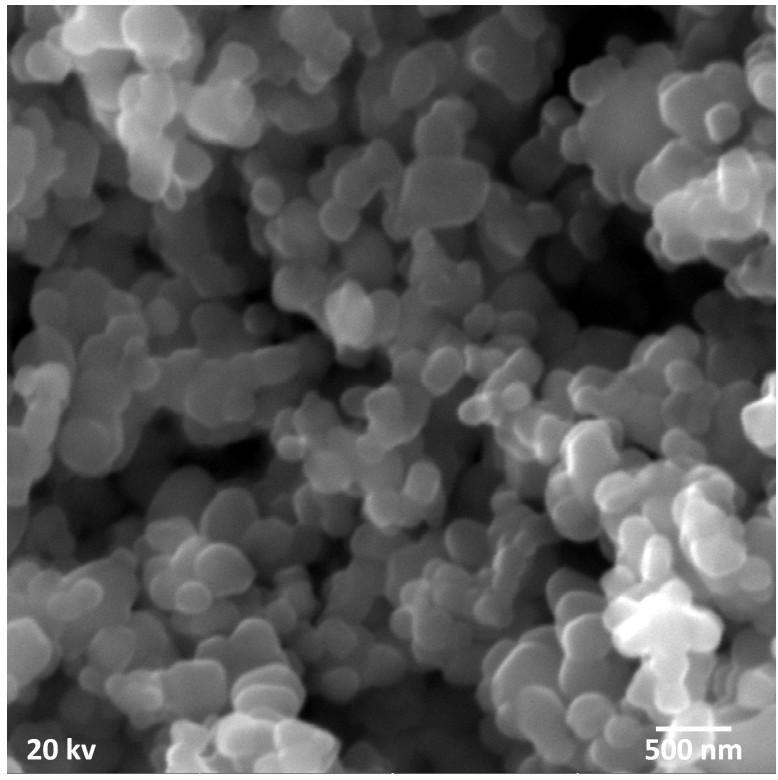

**Figure 1** Scanning electron microscopy of aloe vera based titanium dioxide nanoparticles depicting clustered spherical shape.

## Strength and mineral-content evaluation
### Microhardness evaluation of tooth enamel

*Post hoc* Tukey test revealed that there was a linear pattern increase in the micro-hardness of all the enamel specimens belonging to different age groups with the progression in age before the bleaching with the aloe vera based novel $TiO_2$ -Tooth bleaching gel. The mean Vickers micro-hardness value for the enamel specimen of 20–29 years age group was minimum ($320.35 \pm 0.54$) which increased with age and became ($333.30 \pm 0.41$) in the enamel specimen of 30–39 years age group and ($345.60 \pm 0.67$) in the enamel specimen of 40–49 years age group before the bleaching. The Vickers micro-hardness for the enamel specimen of $\geq 50$ years age group was found to be maximum ($358.70 \pm 0.27$) before the bleaching which was significant ($p$ value $= 0.001$). After carrying out the bleaching procedure, there was a linear pattern decrease in the mean Vickers micro-hardness of all the enamel specimens belonging to different age groups but it was not significant ($p$ value $> 0.05$). The reduction in the mean Vickers micro-hardness for the enamel specimen belonging to 20–29 years age group was ($319.18 \pm 0.52$), ($331.91 \pm 0.63$) in the enamel specimen of 30–39 years age group, ($344.83 \pm 0.63$) in the enamel specimen of 40–49 years age group and finally ($357.55 \pm 0.49$) in the enamel specimen of $\geq 50$ years age group. Thus, mean Vickers micro-hardness levels for enamel specimens belonging to different age
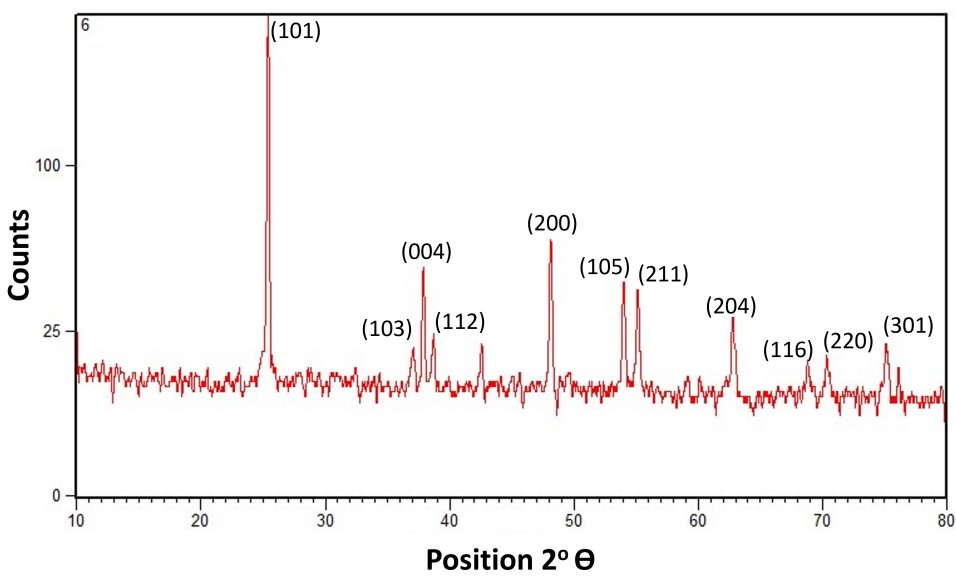

**Figure 2** X-ray diffraction pattern scan of aloe vera based titanium dioxide nanoparticles showing the pure anatase phase with prominent peaks.

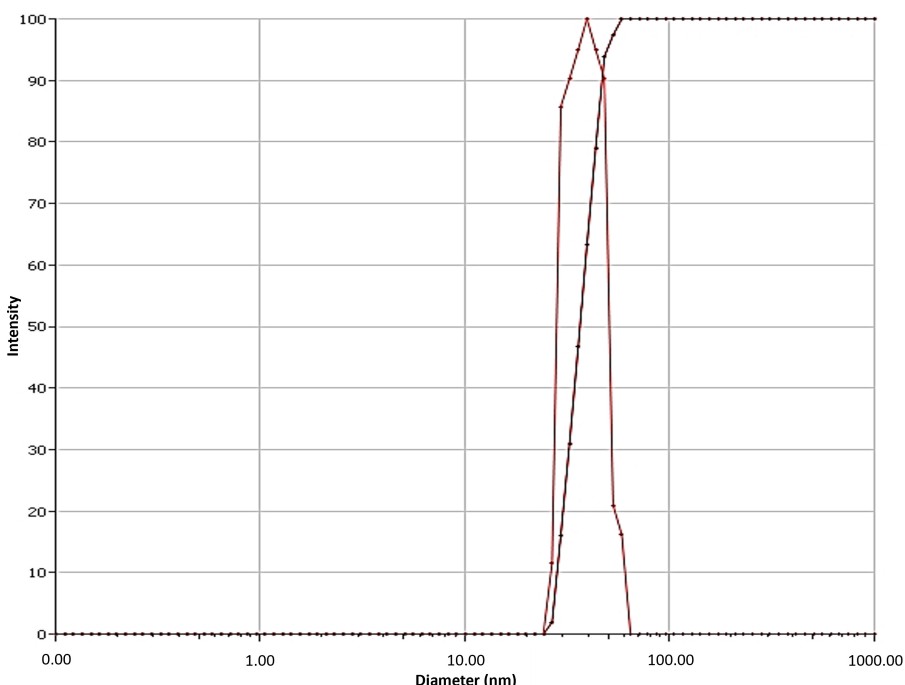

**Figure 3** Dynamic light scattering spectroscopy of aloe vera based titanium dioxide nanoparticles depicting the hydrodynamic particle size.

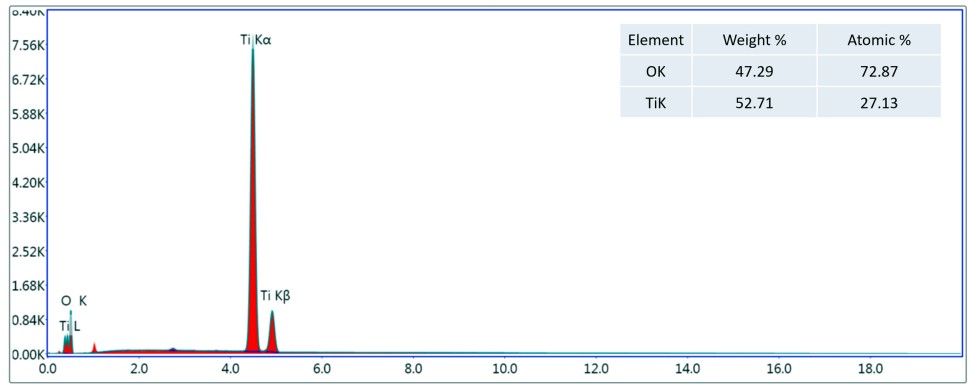

**Figure 4** Element composition of aloe vera based titanium dioxide nanoparticles depicting titanium and oxygen peaks.

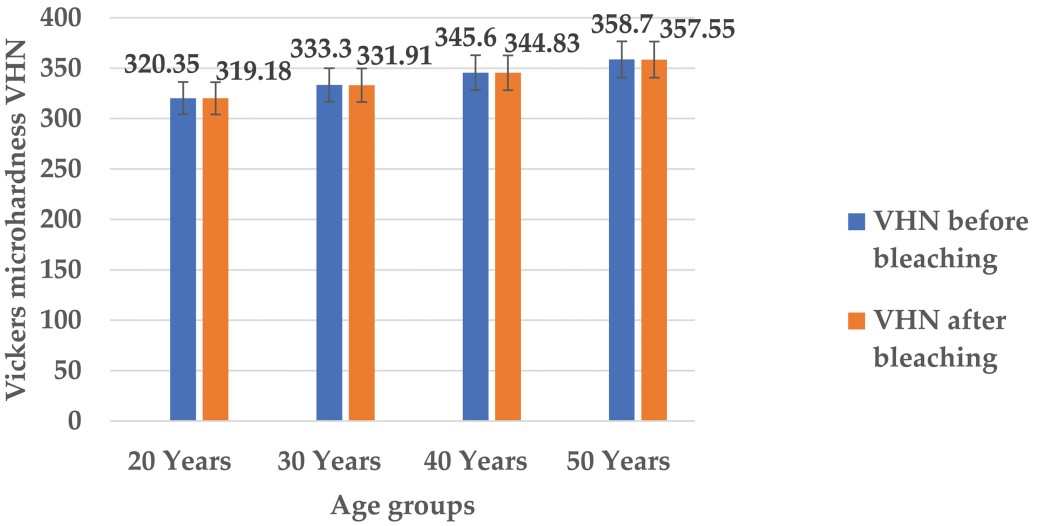

**Figure 5** Mean Vickers micro-hardness levels for enamel specimens of different age groups before and after bleaching.

groups before and after bleaching demonstrated significant difference ($p$ value = 0.001) (Fig. 5, Table 1).

### Mineral-content evaluation of tooth enamel

The EDX scan calculated the minerals in the enamel specimens of all different age groups before and after the bleaching which were found to be sodium (Na), magnesium (Mg), calcium (Ca), and phosphorus (P). There was a linear pattern increase in the mineral-content of all the enamel specimens with the progression in age from 20 to 50 years before the bleaching. The mean values of the basic mineral elements including Na, Mg, Ca, and P were minimum in the enamel specimen of the 20–29 years group, which increased with age in the enamel specimens of 30–39 years and 40–49 years and became maximum in the

**Table 1 Comparison of mean differences of Vickers micro-hardness among enamel specimens belonging to different age groups before and after bleaching.**

| Ser | Comparing Groups | Comparison of Vickers micro-hardness among Different Age Groups before and after bleaching | Mean Difference with S.D and S.E | p Value |
|---|---|---|---|---|
| 1. | Enamel specimen of 20–29 years | Before and after bleaching | $0.193 \pm 0.55$ (0.17) | 0.298 |
| 2. | Enamel specimen of 30–39 years | Before and after bleaching | $0.194 \pm 0.58$ (0.18) | 0.323 |
| 3. | Enamel specimen of 40–49 years | Before and after bleaching | $-0.165 \pm 0.42$ (0.13) | 0.246 |
| 4. | Enamel specimen of $\geq 50$ years | Before and after bleaching | $0.16 \pm 1.62$ (0.51) | 0.755 |

enamel specimen of $\geq 50$ years before bleaching ($p$ value = 0.001). Before bleaching, the mean value of Na, Mg, P, and Ca in the enamel specimen of the 20–29 years group was calculated to be $0.194 \pm 0.05$, $0.176 \pm 0.07$, $13.11 \pm 1.15$, and $25.01 \pm 3.84$, respectively. The mean values of Na, Mg, P, and Ca in the enamel specimen of 30–39 years group was $0.342 \pm 0.04$, $0.370 \pm 0.03$, $15.19 \pm 2.08$, and $27.13 \pm 1.98$, while the mean values of Na, Mg, P, and Ca in the enamel specimen of 40–49 years group was $0.710 \pm 0.09$, $0.690 \pm 0.09$, $17.53 \pm 1.50$, and $30.23 \pm 1.88$. The mean values of Na, Mg, P, and Ca in the enamel specimen of $\geq 50$ years were $0.93 \pm 0.11$, $0.93 \pm 0.09$, $19.79 \pm 1.50$, and $33.98 \pm 1.82$. After bleaching with the aloe vera based novel $TiO_2$-Tooth bleaching gel, there was a prominent decrease in the mineral-content of all the enamel specimens belonging to different age groups but it was not statistically significant ($p$ value > 0.05). After bleaching, the mean value of Na, Mg, P, and Ca in the enamel specimen of the 20–29 years group was $0.192 \pm 0.06$, $0.174 \pm 0.06$, $12.80 \pm 1.19$, and $24.65 \pm 4.38$, respectively. The mean values of Na, Mg, P, and Ca in the enamel specimen of 30–39 years group was $0.340 \pm 0.04$, $0.172 \pm 0.07$, $14.79 \pm 2.47$, and $26.89 \pm 2.31$, whereas the mean values of Na, Mg, P, and Ca in the enamel specimen of 40–49 years group was $0.70 \pm 0.10$, $0.68 \pm 0.09$, $17.22 \pm 2.57$, and $29.91 \pm 2.04$. The mean values of Na, Mg, P, and Ca in the enamel specimen of the $\geq 50$ years group were $0.92 + 0.09$, $0.969 + 0.085$, $19.44 \pm 2.16$, and $33.75 \pm 1.73$, which confirmed insignificant and negligible reduction in mineral content of all enamel specimens. Thus, mean levels of Na, Mg, Ca and P in enamel specimens belonging to different age groups before and after bleaching demonstrated insignificant difference ($p$ value > 0.05) (Fig. 6, Table 2). Moreover, the inter-group reduction in the mineral elements (Na, Mg, Ca, P) calculated in all the enamel specimens belonging to different age groups after bleaching with the novel bleaching gel was minimum and insignificant ($p$ value > 0.05) (Table 2).

## DISCUSSION

Tooth bleaching has become a vital and basic requirement nowadays because the personality and confidence of any individual is highly dependent on the color of the tooth but reduction in hardness strength and elemental composition of tooth enamel is a limitation (*Pinto et al., 2004*; *Alqahtani, 2014*). The advent of nanotechnology has been taken as potent solution for several problems where utilization of titanium dioxide nanoparticles has gained attention because of its prominent characteristics (*Khurshid et al., 2015*; *Zafar et*

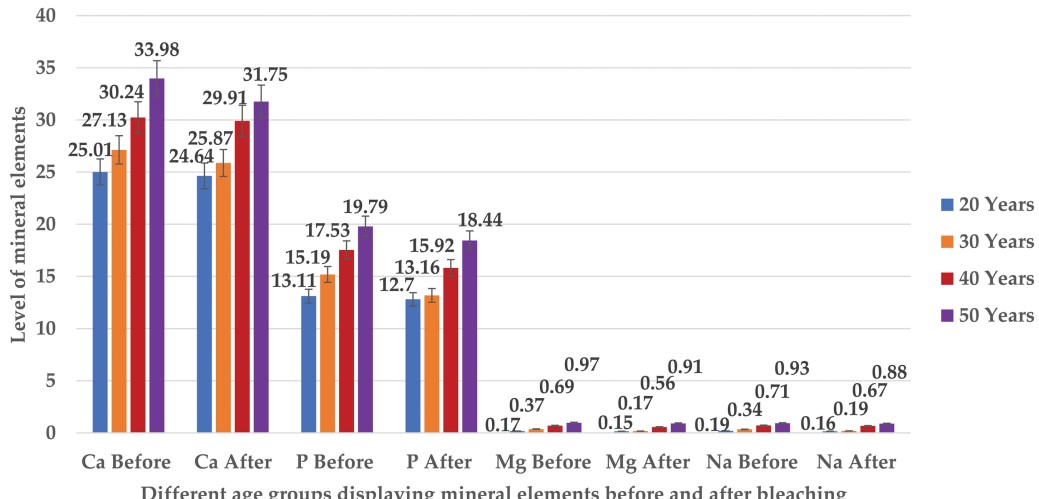

**Figure 6** Bar graph showing the mean level of Na, Mg, Ca and P in enamel specimens of different age groups before and after bleaching.

**Table 2 Comparison of mineral elements among enamel specimens belonging to different age groups before and after bleaching with standard error (SE).**

| Ser | Comparing Age Groups | Comparison of Mineral Elements among Different Age Groups before and after bleaching | Mean Difference with S.D and S.E | p Value |
|---|---|---|---|---|
| 1. | Enamel specimen of 20–29 years | Comparison of Ca before and after bleaching | 0.366 ± 1.70 (0.53) | 0.514 |
| | | Comparison of P before and after bleaching | 0.311 ± 0.66 (0.20) | 0.171 |
| | | Comparison of Mg before and after bleaching | 0.002 ± 0.16 (0.005) | 0.716 |
| | | Comparison of Na before and after bleaching | 0.002 ± 0.16 (0.005) | 0.705 |
| 2. | Enamel specimen of 30–39 years | Comparison of Ca before and after bleaching | 0.239 ± 1.68 (0.53) | 0.664 |
| | | Comparison of P before and after bleaching | 0.401 ± 1.60 (0.50) | 0.451 |
| | | Comparison of Mg before and after bleaching | 0.004 ± 0.01 (0.01) | 0.343 |
| | | Comparison of Na before and after bleaching | 0.002 ± 0.01 (0.01) | 0.693 |
| 3. | Enamel specimen of 40–49 years | Comparison of Ca before and after bleaching | 0.327 ± 1.41 (0.44) | 0.482 |
| | | Comparison of P before and after bleaching | 0.312 ± 1.55 (0.49) | 0.542 |
| | | Comparison of Mg before and after bleaching | 0.006 ± 0.01 (0.01) | 0.217 |
| | | Comparison of Na before and after bleaching | 0.003 ± 0.01 (0.01) | 0.576 |
| 4. | Enamel specimen of ≥ 50 years | Comparison of Ca before and after bleaching | 0.235 ± 1.45 (0.46) | 0.622 |
| | | Comparison of P before and after bleaching | 0.351 ± 1.52 (0.48) | 0.484 |
| | | Comparison of Mg before and after bleaching | 0.005 ± 0.01 (0.01) | 0.397 |
| | | Comparison of Na before and after bleaching | 0.005 ± 0.015 (0.01) | 0.343 |

*al., 2017*), highly biocompatible nature (*Mansoor et al., 2022a*; *Mansoor et al., 2024b*) and strong antimicrobial potency (*Mansoor et al., 2023b*; *Mansoor et al., 2023a*). The novel titanium dioxide nanoparticles synthesized using the aloe vera plant in the current study were beneficial enough to prevent declination of post-bleaching Vickers micro-hardness and mineral content of all the enamel specimens belonging to different age groups. This

might have become possible due to highly stable and strong nature of these titanium dioxide nanoparticles prepared at very low temperatures and pressure. In addition, their synthesis involved the usage of pure biomolecules available in aloe vera plant as reducing and capping agent only. The magnified strength and stability of these nanoparticles could be the only suitable reason for preventing the reduction in Vickers micro-hardness and mineral content of all bleached enamel specimens belonging to different age groups of this study.

Several factors play a crucial role in determining the distinct properties of nanoparticles such as: temperature, pressure, capping, reducing agents used during synthesis, and physicochemical properties acquired after synthesis, such as size, shape, composition, and phase forms. These characteristics play a key role in either making them advantageous or disadvantageous (*Ingale & Chaudhari, 2013*). The ultra nano scaled particles possessed spherical shape (Fig. 1), ideal size (Fig. 2), anatase phase (Fig. 3) and pure elemental composition (Fig. 4), of titanium dioxide nanoparticles obtained in current study might have contributed to their best possible qualities. That might have become possible because of presence of pure and safe 200 bioactive compounds in composition of aloe vera plant (*Kar & Bera, 2018*). The presence of these large amounts of pure bioactive molecules in aloe vera plant might have contributed in the production of highly strong and stable titanium dioxide nanoparticles altogether with ideal size, shape, composition, and phase. These specific properties of the titanium dioxide nanoparticles formed in the present study could have become the sole reason for prohibiting deduction in Vickers micro-hardness (Fig. 5) and mineral content (Fig. 6) of all enamel specimens belonging to different age groups even after bleaching.

Although, currently published study narrated the enhanced bleaching effect on extracted human teeth with minimal effects on their surface roughness and morphology after using titanium oxide induced gel (*Mansoor et al., 2024a*). The previously published literature has already proved that mineral content and Vickers micro-hardness of normal human tooth enamel has been compromised by conventional bleaching agents containing hydrogen peroxide as a result of amount of bleaching agent used, its application time, pH, and concentration. These factors have adversely affected surface roughness, micro-hardness, and mineral content of human tooth (*De Carvalho et al., 2020*; *Vieira et al., 2020*). The utilization of conventional bleaching agent with 35% and 38% hydrogen peroxide has displayed reduction in micro-hardness and mineral content especially calcium and phosphate contents of tooth enamel (*Tezel et al., 2007*), that might have occurred because of lack of stability in hydrogen peroxide (*Broughton, Wentworth & Laing, 1947*). This might have been possible due to highly volatile nature of hydrogen peroxide used in conventional bleaching gels and agents. Additionally, it could have occurred because of chemical interaction with the enamel substrate that might have caused mineral loss. Novel titanium dioxide nanoparticles based bleaching gel fabricated in this study might have prevented hydrogen peroxide from getting unstable in nature. The plausible explanation for this ideal behavior might be small size, spherical shape, pure reactive anatase phase and elemental composition of titanium dioxide nanoparticles produced by green synthesis. These advanced features of novel nanoparticles could have possibly allowed easy penetration

of increased amounts of hydrogen peroxide in tooth surfaces thereby, hindering its conversion in low stability phase form. Additionally, this might have prevented any sort of deterioration in mineral content and loss of micro-hardness from the surfaces of bleached enamel specimens.

The hydrogen peroxide present in bleaching gel undergoes a strong chemical reaction with the stained molecules present in the tooth structure (*Maran et al., 2020*). The results of current study attributed to the structural deterioration and mineral declination were not in accordance with the previous literature where structural changes, demineralization, loss of mineral components (Na, Mg, Ca, P), and decrease in the micro-hardness were significantly observed as a side effect of incorporating conventional tooth bleaching gel containing 36% hydrogen peroxide only (*Maran et al., 2020*; *Grobler, Senekal & Laubscher, 1990*). That could have occurred because of the release of the highly reactive peroxide and superoxide ions from the conventional bleaching gel containing 36% hydrogen peroxide during the oxidation process taking place at the time of bleaching procedure. These free radicals might have degraded the organic matter of the tooth structure especially the loss of mineral elements from its composition and reduction in its micro-hardness (*Hegedüs et al., 1999*). The alterations in the surface morphology and texture of the bleached tooth, are responsible for causing the pulp damage and tooth sensitivity as well (*Soares et al., 2015*). The anatase phase of the novel titanium dioxide nanoparticles is chemically active and stable which might have allowed the interrupted access and roaming of the free radicals in the matrix of the tooth's structure belonging to all the age groups without degrading its micro-hardness and mineral contents, especially Na, Mg, Ca, and P. This controlled flow of free radicals was initiated by the anatase phase of these novel nanoparticles fabricated might have activated the process of bleaching in the tooth enamel without harming its micro-hardness and mineral content. Still, future studies are required to investigate the color changes in the bleached tooth enamel both *in-vivo* and *ex-vivo* to incorporate these nanoparticles in commercial bleaching products without any fear of any side effects.

The aging process has been taken as an important contributing factor in adversely affecting the color, mechanical strength, and organic matrix of the tooth as far as bleaching is concerned. The human tooth enamel of the younger individuals has been found to be more permeable having open tunnels in its organic matrix as compared to the human tooth enamel of the older people. The permeability has been observed to regresses with the shrinking of the tunnels as the age advances from the young to the old (*Kunin, Evdokimova & Moiseeva, 2015*). This shows that younger tooth enamel undergoes more quicker reduction in the micro-hardness and minerals in comparison to the older tooth enamel. Therefore, *Park et al. (2008)* has confirmed the increase of about 12.0%–16.0% in the hardness of the human teeth with aging. This has proved that hardness of the old tooth is much greater than the young tooth (*Park et al., 2008*). Before bleaching with the novel titanium dioxide nanoparticles based gel, the current study has demonstrated a linear pattern increase in hardness and mineral elements (Na, Mg, Ca, P) with the systematic progression in the age. Healthy tooth enamel of 20–29 years group has displayed minimum micro-hardness values and mineral elements (Na, Mg, Ca, P) as compared to the old tooth enamel of ≥50 years group that has exhibited the maximum micro-hardness values and

elements (Na, Mg, Ca, P) before the bleaching (Figs. 5 and 6). After the bleaching, the reduction in the Vickers micro-hardness values and mineral content of the young tooth enamel belonging to the 20–29 years group is slightly more when compared with the old tooth enamel of the ≥50 years group where it was comparatively less. This might have become possible as a result of the open tunnels in the younger tooth enamel of 20–29 years group that has revealed enhanced permeability. This permeability could have allowed the easy and quick ingress of the high concentration of hydrogen peroxide from the novel bleaching gel. As a result, this could have reduced their micro-hardness and mineral content to greater extent in comparison to the tooth enamel of ≥50 years group. This reduction in the mineral content and micro-hardness of the enamel specimens belonging to all the age groups in the current study was negligible that might have occurred because of the incorporation of the stable titanium dioxide nanoparticles. These nanoparticles might have allowed the ingress of a high concentration of hydrogen peroxide from the novel bleaching gel in a well organized and controlled manner that could have prevented the loss of minerals and micro-hardness declination from the bleached enamel specimens belonging to different age groups utilized in this study. Future clinical trials and *in vivo* investigations regarding the induction of nanoparticles in the bleaching gels are required to overcome the complications of teeth sensitivity and pain associated with the bleaching procedures.

## CONCLUSIONS

The green synthesis technique used to synthesize titanium dioxide nanoparticles resulted in nanoscaled small size, spherical shape, 100% anatase phase, and highly pure quality of titanium dioxide nanoparticles. The novel $TiO_2$-bleaching gel prepared from the aloe vera plant was strong and stable enough to prevent the micro-hardness declination and mineral content loss, especially magnesium, sodium, phosphate, and calcium from organic matrix of tooth enamel belonging to different age groups involved in this study. This study concluded that titanium dioxide nanoparticles prepared *via* green route were quite effective for their utilization as naturally derived bleaching gels in clinical dentistry.

## LIMITATION OF STUDY

Future studies are required for better understanding of the bleaching procedure with the nanoparticles because the small sample size and *in-vitro* investigation on the extracted human teeth are the limitations of our study where further research can overcome these constraints.

## ACKNOWLEDGEMENTS

The authors acknowledge Department of Materials, National Institute of Lasers and Optronics (NILOPE, Islamabad) and Pakistan Institute of Engineering and Applied Sciences, (PIEAS, Islamabad) for the use of their laboratories for this work.

### Funding

This work was supported by the Deputyship for Research and Innovation, Ministry of Education, Kingdom of Saudi Arabia. This APC was supported by the Deanship of Scientific Research, Vice Presidency for Graduate Studies and Scientific Research, King Faisal University, Saudi Arabia (Grant No. KFU241396). The funders had no role in study design, data collection and analysis, decision to publish, or preparation of the manuscript.

### Grant Disclosures

The following grant information was disclosed by the authors:
The Deputyship for Research and Innovation, Ministry of Education, Kingdom of Saudi Arabia.
The Deanship of Scientific Research, Vice Presidency for Graduate Studies and Scientific Research, King Faisal University, Saudi Arabia: KFU241396.

### Competing Interests

The authors declare there are no competing interests.

### Author Contributions

- Afsheen Mansoor conceived and designed the experiments, performed the experiments, analyzed the data, prepared figures and/or tables, authored or reviewed drafts of the article, and approved the final draft.
- Emaan Mansoor conceived and designed the experiments, performed the experiments, analyzed the data, prepared figures and/or tables, authored or reviewed drafts of the article, and approved the final draft.
- Atta Ullah Shah performed the experiments, analyzed the data, prepared figures and/or tables, and approved the final draft.
- Uzma Asjad performed the experiments, analyzed the data, prepared figures and/or tables, and approved the final draft.
- Zohaib Khurshid analyzed the data, authored or reviewed drafts of the article, and approved the final draft.
- Amir Isam Omer Ibrahim analyzed the data, authored or reviewed drafts of the article, and approved the final draft.

### Human Ethics

The following information was supplied relating to ethical approvals (*i.e.*, approving body and any reference numbers):

The School of Dentistry, Shaheed Zulfiqar Ali Bhutto Medical University, Islamabad, Pakistan has approved this study (SOD/EBR/2022/24)

### Data Availability

The raw data are available in the Supplemental Files.

## Supplemental Information

Supplemental information for this article can be found online at http://dx.doi.org/10.7717/peerj.17779#supplemental-information.

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
