# Peer review of "Role of the novel aloe vera-based titanium dioxide bleaching gel on the strength and mineral content of the human tooth enamel with respect to age"

_PeerJ, doi:10.7717/peerj.17779_

## Round 0.1 · original submission · Major Revisions

The topic is interesting. Kindly address all the suggestions made by the reviewers.

**Language Note:** The review process has identified that the English language must be improved. PeerJ can provide language editing services - please contact us at [email protected] for pricing (be sure to provide your manuscript number and title). Alternatively, you should make your own arrangements to improve the language quality and provide details in your response letter. – PeerJ Staff

·

Basic reporting

There are several grammatical errors throughout the text. The authors may benefit from using AI supported proof reading tools such as Grammarly.

Please provide reference for line 121

Experimental design

Was there any formal sample size calculation? If not please state that the sample size chosen was a convenience sample and include power of the results.

Validity of the findings

Was there a control group that was bleached using a conventional 35% hydrogen peroxide gel without the TiO2? It is difficult to make any meaningful conclusions withput a control group. Please include data on a control group.

Please verify if reference 25 is correctly placed in the discussion.

The statement in line 294-296 that reduction in hardness is the major limitation of conventional bleaching is misleading. Please correct the statement to "is a limitation"

One of the conclusions is that " This 434
study concluded that Titanium dioxide nanoparticles prepared via green route were 435
pretty much effective in promoting the aesthetics without compromising the strength 436
and mineral-content of the human teeth in all the age groups." Please exclude 'aesthetics' as this was not assessed in this study.

Reviewer 2 ·

Basic reporting

This is very well written. Thank you for providing specific explanations and having great English.

Experimental design

The design is well written, but there is no comparison with a conventional bleaching gel to demonstrate the advantages.

Validity of the findings

-Sample sizes are quite small (n=7 per group).
-Please provide an analysis of the color changes in the enamel. Maybe evaluate color change in enamel as an added outcome measure.

Reviewer 3 ·

Basic reporting

Dear Authors,
I have reviewed the manuscript titled "Role of the novel Aloe Vera-based Titanium dioxide bleaching gel on the strength and mineral content of the human tooth enamel with respect to age" submitted to PeerJ. Overall, the study aims to evaluate the novel titanium dioxide bleaching gel's effects on tooth enamel belonging to different age groups. The topic is interesting and addresses an important issue. However, there are some major issues that need to be addressed before considering the manuscript for publication.

Language and structure:
The language needs improvement to ensure clarity, as there are multiple instances of awkward phrasing and unclear sentences. I suggest having a fluent English editor thoroughly proofread the manuscript. Additionally, the introduction requires expansion to establish more context and significance. Certain sections like the results and patents also need better structuring.

Figures and tables: Figures are relevant but require higher resolution/quality and more descriptive legends for clarity. Similarly, tables need better formatting and presentation.

References: The references appear extensive but have inconsistent formatting at places. Please ensure references adhere to PeerJ citation style requirements. There is also a high proportion of self-citations, which should be limited.

Methods: More details are needed on ethical approvals, tooth sample collection protocols, sample size rationalization, and standardization of experimental procedures. The statistical analysis also requires more elaboration.

Results: This section needs better organization under descriptive sub-headings. There is also scope to summarize some repetitive data more concisely via tabulation. Discussion of limitations is currently inadequate.

Conclusions: The conclusions highlight the promising effects of the bleaching gel but overstate it as definitive when study constraints like sample size and in-vitro nature limit generalizability. I suggest toning down the conclusiveness and calling for more research.

In summary, I think the manuscript has merit but requires major revisions, as outlined above, to address issues related to the presentation, completeness of reporting, data analysis, and balanced interpretation sufficiently. Doing so can better convey the relevance of this work and allow reconsideration. I hope you find these comments helpful to strengthen your paper.

Kind regards,

Experimental design

No comment

Validity of the findings

No comment

Additional comments

No comment

---

## Round 0.2 · Minor Revisions

Please revise the manuscript in accordance with the suggestions of the reviewers.

Reviewer 2 ·

Basic reporting

The introduction provides a good overview, but more detail is needed. The authors should expand on their rationale and should improve their English.

Experimental design

Well done.

Validity of the findings

The results section is comprehensive and well-described.

The conclusions should focus on the primary research question and implications, rather than being a broad summary.

The limitations, such as the relatively small sample size and the in vitro nature of the experiments, should be acknowledged, and suggestions for future studies should be provided.

---

## Round 0.3 · accepted · Accept

The authors have incorporated all of the suggestions provided by the reviewers.